# Patch2Self: Denoising Diffusion MRI with Self-Supervised Learning

**Shreyas Fadnavis**[1][*], **Joshua Batson**[2][†], **Eleftherios Garyfallidis**[1][†]
[1]Indiana University Bloomington, [2]CZ Biohub

## Abstract

Diffusion-weighted magnetic resonance imaging (DWI) is the only noninvasive method for quantifying microstructure and reconstructing white-matter pathways in the living human brain. Fluctuations from multiple sources create significant additive noise in DWI data which must be suppressed before subsequent microstructure analysis. We introduce a self-supervised learning method for denoising DWI data, Patch2Self, which uses the entire volume to learn a full-rank locally linear denoiser for that volume. By taking advantage of the oversampled $q$-space of DWI data, Patch2Self can separate structure from noise without requiring an explicit model for either. We demonstrate the effectiveness of Patch2Self via quantitative and qualitative improvements in microstructure modeling, tracking (via fiber bundle coherency) and model estimation relative to other unsupervised methods on real and simulated data.

## 1  Introduction

Diffusion Weighted Magnetic Resonance Imaging (DWI) [3] is a powerful method for measuring tissue microstructure [33, 34]. Estimates of the diffusion signal in each voxel can be integrated using mathematical models to reconstruct the white matter pathways in the brain. The fidelity of those inferred structures is limited by the substantial noise present in DWI acquisitions, due to numerous factors including thermal fluctuations. With new acquisition schemes or diffusion-encoding strategies, the sources and distribution of the noise can vary, making it difficult to model and remove. The noise confounds both qualitative (visual) and quantitative (microstructure and tractography) analysis. Denoising is therefore a vital processing step for DWI data prior to anatomical inference.

DWI data consist of many 3D acquisitions, in which diffusion along different gradient directions is measured. Simple models of Gaussian diffusion are parametrized by a six-dimensional tensor, for which six measurements would be sufficient, but as each voxel may contain an assortment of tissue microstructure with different properties, many more gradient directions are often acquired. While each of these acquired volumes may be quite noisy, the fact that the same structures are represented in each offers the potential for significant denoising.

The first class of denoising methods used for DWI data were extensions of techniques developed for 2D images, such as non-local means (NL-means [9] and its variants [36, 8]), total variation norm minimization [24], cosine transform filtering [30], empirical Bayes [2] and correlation based joint filtering [47]. Other methods take more direct advantage of the fact that DWI measurements have a special 4D structure, representing many acquisitions of the same 3D volume at different b-values and in different gradient directions. Assuming that small spatial structures are more-or-less consistent across these measurements, these methods project to a local low-rank approximation of the data [37, 31]. The top performing methods are overcomplete Local-PCA (LPCA) [31] and its Marchenko-Pastur extension [48]. The current state-of-the-art unsupervised method for denoising DWI is the Marchenko-Pastur PCA, which handles the choice of rank in a principled way by thresholding based on the eigenspectrum of the expected noise covariance matrix. Note that Marchenko-Pastur PCA,

like the classical total variation norm and NL-means methods as well, requires a noise model to do the denoising, either as an explicit standard deviation and covariance as in LPCA, or implicitly in the choice of a noise correction method [25, 44].

We propose a self-supervised denoising method for DWI data that incorporates the key features of successful prior methods while removing the requirement to select or calibrate a noise model. Our method, Patch2Self, learns locally linear relationships between different acquisition volumes on small spatial patches. This regression framework satisfies the $\mathcal{J}$-invariance property described in [4], which, as long as the noise in different acquisitions is statistically independent, will guarantee denoising performance. With thorough comparisons on real and simulated data, we show that Patch2Self outperforms other unsupervised methods for denoising DWI at visual and modeling tasks.

## 2    Self-Supervised Local Low Rank Approximation

### 2.1    Approach and Related Work

Among different approaches to denoising, methods based on low-rank approximations have given the most promising results for DWI denoising. Representing the data as low-rank requires accurate estimation of a threshold to truncate the basis set, typically constructed via eigen-decomposition of the data. The only unsupervised approach for doing so in the case of DWI makes use of random-matrix-theory based Marchenko-Pastur distribution to classify the eigenvalues pertaining to noise [48]. Extending the idea of LPCA, it computes a universal signature of noise, in the PCA domain. However, this is constrained by the variability in denoising caused by local patch-sizes and assumption of the noise being homoscedastic. Patch2Self is the first of its kind approach to DWI denoising that leverages the following two properties:

#### 2.1.1    Statistical Independence of Noise

Since noise is perceived as random fluctuations in the signal measurements, one can assume that the noise in one measurement is independent of another. Using this, [29] showed that given two noisy independent images, we could use one measurement to denoise another by posing denoising as a prediction problem. Expanding this idea, [4] laid out a theoretically grounded notion of self-supervision for denoising signals from the same corrupted image/ signal measurement. Leveraging this idea of statistical independence, different approaches such as [26] and [27] have shown competitive performance with varying approximation settings. However, most of these approaches tackle 2D images typically via CNN-based deep learning methods with the motive of improving the visual quality of the image (i.e., they do not account for modeling the signal for scientific analysis like DWI). These approaches are not feasible in 4D DWI data, as the denoising needs to be unsupervised, fast and clinically viable for downstream image-analysis. In Patch2Self, we delineate how one can extrapolate the notion of noise independence purely via patches in a 4D volumetric setting via a regression framework.

#### 2.1.2    Patches and Local Matrix Approximations

Patch-based self-supervision has been used to learn representations that are invariant to distortions [13, 12], for learning relations between patches [11], for filling in missing data (i.e. image in-painting) [38], etc. Patch2Self abides by a similar patch-based approach where we learn an underlying clean signal representation that is invariant to random fluctuations in the observed signal. Inspired by the local matrix approximation works presented in [28, 7], we formulate a global estimator per 3D volume of the 4D data by training on local patches sampled from the remaining volumes. This estimator function, thus has access to local and non-local information to learn the mapping between corrupted signal and true signal, similar to dictionary learning [18, 42, 45, 20, 6] and non-local block matching [10]. Due to the self-supervised formulation, Patch2Self can be viewed as a non-parametric method that regresses over patches from all other volumes except from the one that is held-out for denoising. Similar to [43], our experiments demonstrate that a simplistic linear-regression model can be used to denoise noisy matrices using $p$-neighbourhoods and a class of $\mathcal{J}$-invariant functions.

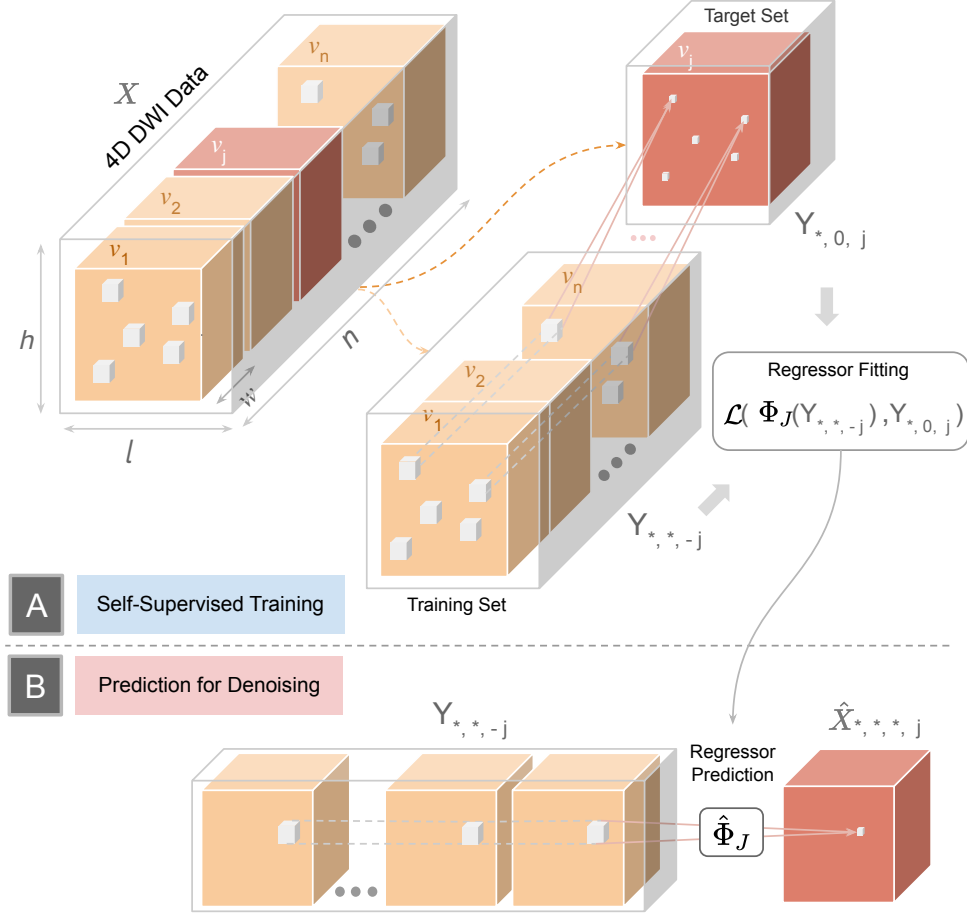

Figure 1: Depicts the workflow of Patch2Self in two phases: **(A)** Is the self-supervised training phase where the 4D DWI data is split into the training $Y_{*,*,-j}$ and target $Y_{*,0,j}$ sets. $p$-neighbourhoods are extracted from each 3D volume from both $Y_{*,*,-j}$ and $Y_{*,0,j}$. $\Phi_J$ is the learnt mapping by regressing over $p$-neighbourhoods of $Y_{*,*,-j}$ to estimate $Y_{*,0,j}$. **(B)** Depicts the voxel-by-voxel denoising phase where $\hat{\Phi}_J$ predicts the denoised volume $\hat{X}_{*,*,*,j}$ from $Y_{*,*,-j}$.

## 2.2 Denoising via Self-Supervised Local Approximations

**Extracting 3D patches** In the first phase of Patch2Self, we extract a $p$-neighbourhood for each voxel from the 4D DWI data. To do so, we construct a 3D block of radius $p$ around each voxel, resulting in a local $p$-neighbourhood of dimension $p \times p \times p$. Therefore, if the 4D DWI has $n$ volumes $\{v_1, \ldots, v_n\}$ (each volume corresponding to a different gradient direction) and each 3D volume has $m$ voxels (see Fig. 1), after extracting the $p$-neighbourhoods, we get a $m \times p \times p \times p \times n$ tensor. Next, we flatten this this tensor along the $p^{th}$-dimension to obtain a representation: $m \times (p^3 \times n)$. Thus, we have transformed the data from the original 4D space to obtain $m$ samples of $p^3 \times n$ dimensional 2D feature matrices, which we use to train the denoiser.

**Self-Supervised Regression** In the second phase, using the $p$-neighbourhoods, Patch2Self reformulates the problem of denoising with a predictive approach. The goal is to iterate and denoise each 3D volume of the noisy 4D DWI data ($X$) using the following training and prediction phases:

(i) Training: To denoise a particular volume, $v_j$, we train the a regression function $\Phi_J$ using $p$-neighbourhoods of the voxels denoted by the set $Y$. From the first phase, $Y$ is a set containing $m$ training samples with dimension: $p^3 \times n$. Next, we hold out the dimension corresponding to volume $v_j$ from each of the $p$-neighbourhoods and use it as a target for training the regressor function $\Phi_J$ (shown in Fig. 1A). Therefore our training set $Y_{*,*,-j}$ has dimension: $m \times p^3 \times (n-1)$, where $j$ indexes the held out dimension of the $p$-neighbourhoods set. Using the regressor function $\Phi_J$, we

use the training set $Y_{*,*,-j}$ to only predict the center voxel of the set of $p$-neighbourhoods in the corresponding target set of dimension $Y_{*,0,-j}$. The target set, is therefore only an $m$-dimensional vector of the center voxels of the corresponding $p$-neighbourhoods of volume $v_j$. In summary, we use the localized spatial neighbourhood information around each voxel of the set of volumes $v_{-j}$, to train $\Phi_J$ for approximating the center voxel of the target volume $v_j$. To do so, we propose minimizing the self-supervised loss over the described $p$-neighbourhood sets as follows:

$$\mathcal{L}(\Phi_J) = \mathbb{E}\|\Phi_J(Y_{*,*,-j}) - Y_{*,0,-j}\|^2 \tag{1}$$

Where, $\Phi_J : \mathbb{R}^{p^3 \times n} \mapsto \mathbb{R}^1$, is trained on $m$ samples of $p$-neighbourhoods.

(ii) Predict: After training for $m$ samples, we have now constructed a $\mathcal{J}$-invariant regressor $\hat{\Phi}_J$ that can be used to denoise the held out volume $v_j$. To do so, $p$-neighbourhoods from the set $Y_{*,*,-j}$ are simply fed into $\hat{\Phi}_J$ to obtain the denoised $p$-neighbourhoods corresponding to the denoised volume $\hat{Y}_{*,0,-j}$. After doing so, for each $j \in \{1 \dots n\}$, we unravel the $p$-neighbourhoods for each volume $v_j \in \{v_1 \dots v_n\}$ (in Fig. 1 as $\hat{X}_{*,*,*,j}$) and append them to obtain the denoised 4D DWI data $\hat{X}$.

$\mathcal{J}$-**Invariance** The reason one might expect the regressors learned using the self-supervised loss above to be effective denoisers is the theory of $\mathcal{J}$-invariance introduced in [4]. Consider the partition of the data into volumes, $\mathcal{J} = \{v_1, \dots, v_n\}$. If the noise in each volume is independent from the noise in each other volume, and

**Algorithm:** Patch2Self

Input 4D data $X$ of dimension $l \times w \times h \times n$
**for** volume $j = 1, 2, \dots n$ **do** [where $n$ is the number of volumes]
    **for** voxel $k = 1, 2, \dots m$ **do** [where $m = lwh$ is the number of voxels]
        Extract a $p \times p \times p$ neighbourhood of voxel $k$
    Flatten and concatenate the $p$-neighbourhood of each voxel into a feature vector of length $p^3 \times n$.
Stack feature vectors into a matrix of size $m \times (p^3 \times n)$.
**for** volume $j = 1, 2, \dots n$ **do**
    Hold-out features from volume $j$ to get a feature matrix $Y_{*,*,-j}$ of dimension $m \times p^3 \times n - 1$
    Select the central pixels from volume $j$ to get a target vector $Y_{*,0,j}$ of dimension $m$.
    Train a linear regressor $\Phi : Y_{*,*,-j} \mapsto Y_{*,0,j}$.
    Set the denoised volume $\hat{X}_{*,*,*,j}$ to the unraveled output $\hat{\Phi}(Y_{*,*,-j})$.
**return** Denoised 4D data $\hat{X}$

a denoising function $\Phi$ satisfies the property that the output of $\Phi$ in volume $v_j$ does not depend on the input to $\Phi$ in volume $v_j$, then according to Proposition 1 of [4], the sum over all volumes of the self-supervised losses in equation 1 will in expectation be equal to the ground-truth loss of the denoiser $\Phi$, plus an additive constant. This means that $\mathcal{J}$-invariant functions minimizing the self-supervised loss will also minimize the ground-truth loss. This holds by construction for our denoiser $\Phi = (\Phi_1, \dots, \Phi_n)$. Intuitively, each $\Phi_J$ only has access to the signal present in the volumes other than $v_j$, and since the noise in those volumes is irrelevant for predicting the noise in $v_j$, it will learn to suppress the fluctuations due to noise while preserving the signal. Note that, if linear regression is used to fit each $\Phi_J$, then the final denoiser $\Phi$ is a linear function. Unlike methods which work by thresholding the singular values obtained from a local eigen-decomposition [31, 48], which produce denoised data that are locally low-rank, this mapping $\Phi$ can be full-rank.

**Choice of Regressor:** Any class of regressor can be fit in the above method, from simple linear regression/ordinary least squares to regularized linear models like Ridge and Lasso, to more complex nonlinear models. Our code-base allows for the use of any regression model from [39]. Surprisingly, we found that linear regression performed comparably to the more sophisticated models, and was of course faster to train (see supplement for comparisons).

**Choice of Patch Radius:** To determine the effect of changing the patch radius on denoising accuracy, we compute the Root Mean Squared Error (RMSE) between the ground truth and Patch2Self denoised estimates at SNR 15 (details of simulation in Sec. 4). For patch radius zero and one, we show the effect at different number of volumes as depicted in Fig. 6C. The line-plot trend shows that the difference in the RMSE scores between the two patch radii steadily decreases with an increase in number of volumes. However, with lesser number of volumes, a bigger patch-radius must be used. In the remainder of the text, we use and show results with patch radius zero and linear regressors.

## 3 Evaluation on Real Data

### 3.1 Evaluation on *in-vivo* data

We compare the performance of Patch2Self with Marchenko-Pastur on the Parkinson's Progression Markers Initiative (PPMI) [32], Stanford HARDI [41] and Sherbrooke 3-Shell [16] datasets as shown

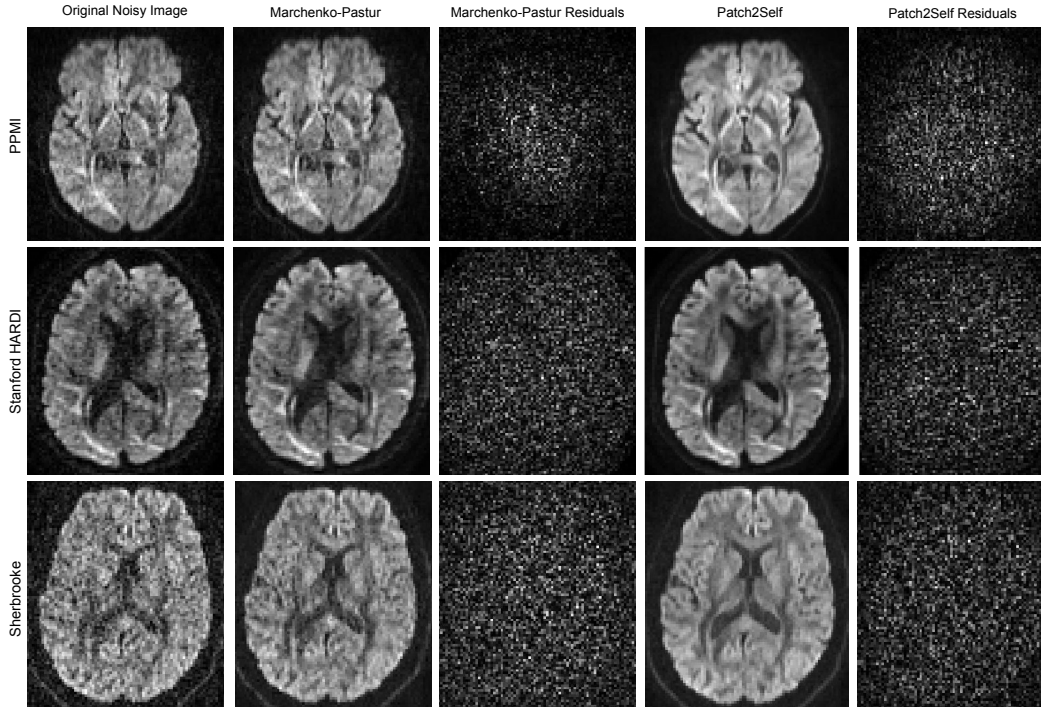

Figure 2: Shows the comparison of denoising on 3 different types of datasets: Parkinson's Progression Markers Initiative (PPMI), Stanford HARDI and Sherbrooke 3-Shell HARDI data. The denoising of Patch2Self is compared against the original noisy image and Marchenko-Pastur denoised data along with their corresponding residuals. Notice that Patch2Self suppresses more noise and also does not show any anatomical structure in the corresponding residual plots.

in Fig. 2. These datasets represent different commonly used acquisition schemes: (1) Single-Shell (PPMI, 65 gradient directions), (2) High-Angular Resolution Diffusion Imaging (Stanford HARDI, 160 gradient directions) and (3) Multi-Shell (Sherbrooke 3-Shell, 193 gradient directions). For each of the datasets, we show the axial slice of a randomly chosen 3D volume and the corresponding residuals (squared differences between the noisy data and the denoised output). Note that both, Marchenko-Pastur and Patch2Self, do not show any anatomical features in the error-residual maps, so it is likely that neither is introducing structural artifacts. Patch2Self produced more visually coherent outputs, which is important as visual inspection is part of clinical diagnosis.

## 3.2 Effect on Tractography

To reconstruct white-matter pathways in the brain, one integrates orientation information of the underlying axonal bundles (streamlines) obtained by decomposing the signal in each voxel using a microstructure model [5, 34]. Noise that corrupts the acquired DWI may impact the tractography results, leading to spurious streamlines generated by the tracking algorithm. We evaluate the effects of denoising on probabilistic tracking [17] using the Fiber Bundle Coherency (FBC) metric [40]. To perform the probabilistic tracking, the data was first fitted with the Constant Solid Angle (CSA) model [1]. The Generalized Fractional Anisotropy (GFA) metric extracted from this fitting was used as a stopping criterion within the probabilistic tracking algorithm. The fiber orientation distribution information required to perform the tracking was obtained from the Constrained Spherical Decon-volution (CSD) [46] model fitted to the same data. In Fig. 3, we show the effect of denoising on tractography for the Optic Radiation (OR) bundle as in [40]. The OR fiber bundle, which connects the visual cortex:V1 (calcarine sulcus) to the lateral geniculate nucleus (LGN), was obtained by selecting a $3 \times 3 \times 3$ Region Of Interest (ROI) using a seeding density of 6. After the streamlines were generated, their coherency was measured with the local FBC algorithm [40, 14]), with *yellow-orange* representing - spurious/incoherent fibers and *red-blue* representing valid/coherent fibers. In Fig, 3, OR

bundle tracked from original/ raw data contains 3114 streamlines, Marchenko-Pastur denoised data [48] contains 2331 streamlines and Patch2Self denoised data contains 1622 streamlines. Patch2Self outperforms Marchenko-Pastur by reducing the number of incoherent streamlines, as can be seen in the *red-blue* (depicting high coherence) coloring in Fig. 3.

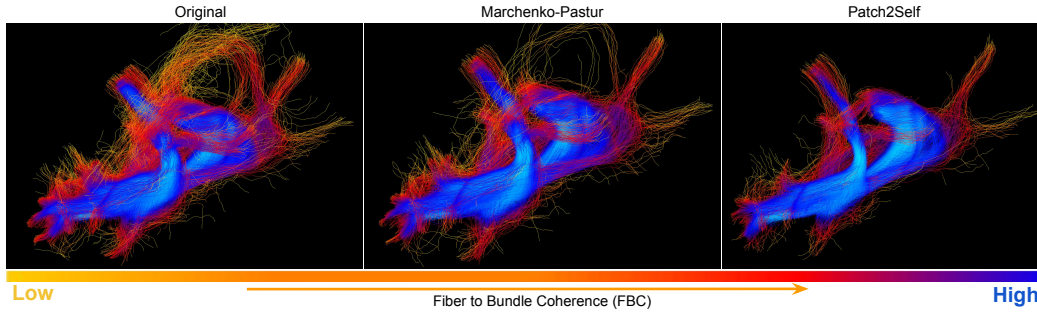

Figure 3: Depicts the Fiber to Bundle Coherency (FBC) density map projected on the streamlines of the optic radiation bundle generated by the probabilistic tracking algorithm. The color of the streamlines depicts the coherency − *yellow* corresponding to incoherent and *blue* corresponding to coherent. Notice that the number of incoherent streamlines present in the original fiber-bundle is reduced after Marchenko-Pastur denoising. Patch2Self denoising further reduces spurious tracts, resulting in a cleaner representation of the fiber bundle.

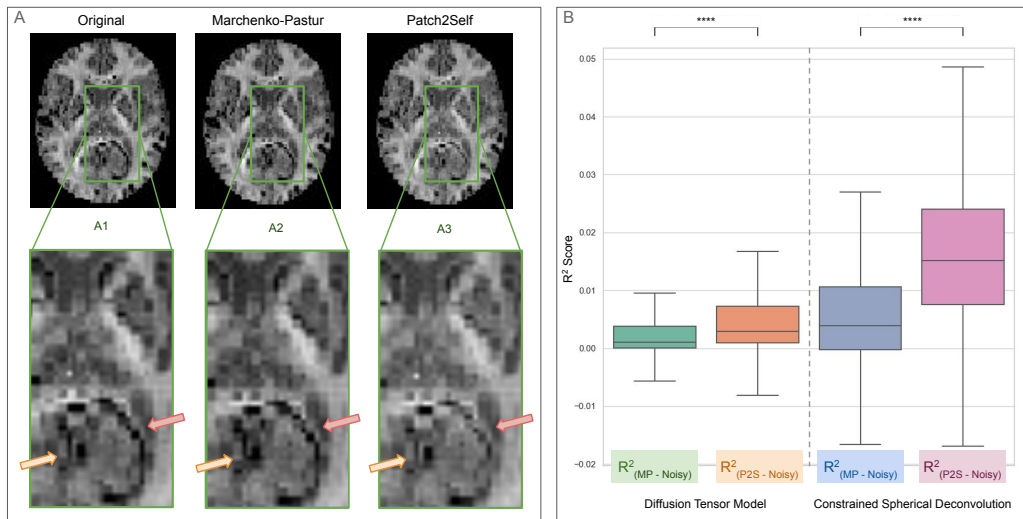

Figure 4: **(A)** Shows the mean kurtosis parameter maps obtained by fitting the DKI model to the axial slice of the CFIN dataset [21]. Notice that Patch2Self **(A3)** alleviates more degenerecies in model estimation (visible as black voxels in the region highlighted with arrows) as compared to noisy (original) data **(A1)** and Marchenko-Pastur **(A2)** denoised data. **(B)** Box-plots quantifying the increase in $R^2$ metric after fitting downstream DTI and CSD models. The $R^2$ improvements in each case are plotted by subtracting the scores of model fitting on noisy data from $R^2$ of fitting each denoised output. Note that the consistency of microstructure model fitting on Patch2Self denoised data is higher than that obtained from Marchenko-Pastur (see 3.3 for details and significance).

## 3.3 Impacts on Microstructure Model Fitting

The domain of microstructure modeling employs either mechanistic or phenomenological approaches to resolve tissue structure at a sub-voxel scale. Fitting these models to the data is a hard inverse problem and often leads to degenerate parameter estimates due to the low SNR of DWI acquisitions [35]. We apply two of the most commonly used diffusion microstructure models, Constrained

Spherical Deconvolution (CSD) [46] and Diffusion Tensor Imaging (DTI) [3], on raw and denoised data. DTI is a simpler model that captures the local diffusion information within each voxel by modeling it in the form of a 6-parameter tensor. CSD is a more complex model using a spherical harmonic representation of the underlying fiber orientation distributions. In order to compare the goodness of each fit, we perform a k-fold cross-validation (CV) [22] at two exemplary voxel locations, corpus callosum (CC), a single-fiber structure, and centrum semiovale (CSO), a crossing-fiber structure. The data is divided into $k = 3$ different subsets for the selected voxels, and data from two folds are used to fit the model, which predicts the data on the held-out fold. The scatter plots of CV predictions against the original data are shown in Fig. 5 for those two voxels. As measured by $R^2$, Patch2Self has a better goodness-of-fit than Marchenko-Pastur by $22\%$ for CC and $65\%$ for CSO. To show that Patch2Self consistently improves model fitting across all voxels, in Fig. 4B we depict the improvement of the $R^2$ metric obtained from the same procedure for the axial slice (4606 voxels) of masked data (using [41] data). This was done by simply subtracting the goodness-of-fit $R^2$ scores of fitting noisy data, from Marchenko-Pastur and Patch2Self denoised data for both CSD and DTI models. Patch2Self shows a significant improvement on both DTI and CSD (two-sided t-test, p < 1e-300, Fig. 4). The Diffusion Kurtosis (DKI) model contrast, uses higher-order moments to quantify the non-gaussianity of the underlying stochastic diffusion process. This can be used to characterize microstructural heterogeneity [23] leading to important biomarkers of axonal fiber density and diffusion tortuosity [15]. Models such as DKI are susceptible to noise and signal fluctuations can often lead to estimation degeneracies. In Fig. 4A, we compare the effects of different denoising algorithms on DKI parameter estimation by visualizing the *mean kurtosis* maps. We make use of the CFIN dataset [21] which was designed to evaluate kurtosis modeling and imaging strategies to depict the effects of denoising. As highlighted by the arrows, Marchenko-Pastur does not add any new artifacts due to noise suppression but also does not help alleviate degeneracies in parameter estimation. Patch2Self reduces the number of degeneracies without adding any artifacts due to denoising.

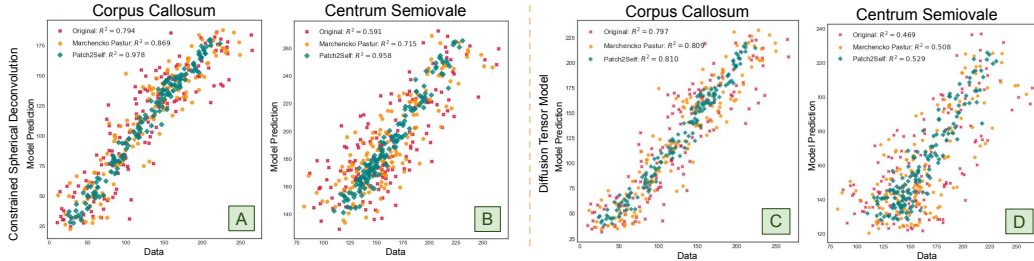

Figure 5: Quantitative comparison of the goodness-of-fit evaluated using a cross-validation approach. (**A**) and (**B**) depict the scatter plots of the model predictions obtained by fitting CSD to voxels in the corpus callosum (CC) and centrum semiovale (CSO) for original (noisy), Marchenko-Pastur (denoised) and Patch2Self (denoised) data. Similarly (**C**) and (**D**) show the scatter plots of predictions obtained from DTI fitting in the same voxel locations. Top-left of each plot shows the $R^2$ metric computed from each model fit on the corresponding data.

## 4 Evaluation on Simulated Data

We begin with whole brain noise-free DWIs simulated from the framework proposed in [19, 49], with 2 image volumes at b-value=0 $s/mm^2$ (i.e., b0), 30 diffusion directions at b-value=1000 $s/mm^2$ (b1000) and 30 directions at b-value=2000 $s/mm^2$ (b2000). The real-world noise distribution was simulated with multi-channel acquisition: a realistic 8-channel coil sensitivity map was used and Gaussian noise was added to the real and imaginary part of each channel of the DWIs respectively [49]. Finally DWIs were combined with sum-of-square coil combination and signal-to-noise ratio (SNR) was calculated in the white-matter of the b=0 image. All together, 5 datasets were simulated: noise-free, SNR= 10, 15, 20, 25 and 30. We take two different approaches of comparing Patch2Self and the performance gains it provides: (1) Compute the mean squared error (MSE) between the denoised data and the ground truth at each SNR. (2) Using the $R^2$ metric of the denoised data against ground truth. The outcomes from both these evaluation strategies have been compared against the raw noisy data and Marchenko-Pastur denoised data. As shown in table 1 the performance gains obtained

by using Patch2Self are substantial in realistic SNR ranges, especially in the 5-20 range common for in-vivo imaging. This can also be seen qualitatively in Fig. 6, where Patch2Self is visibly cleaner than with Marchenko-Pastur[48] at each low SNRs. A scatterplot of ground-truth versus denoised voxel value illustrates this performance gain as well (Fig. 6D). Notice that with an increase in SNR, the performance of Patch2Self improves consistently (see table 1) and does consistently better than the Marchenko-Pastur method (evaluated via MSE and $R^2$ metrics).

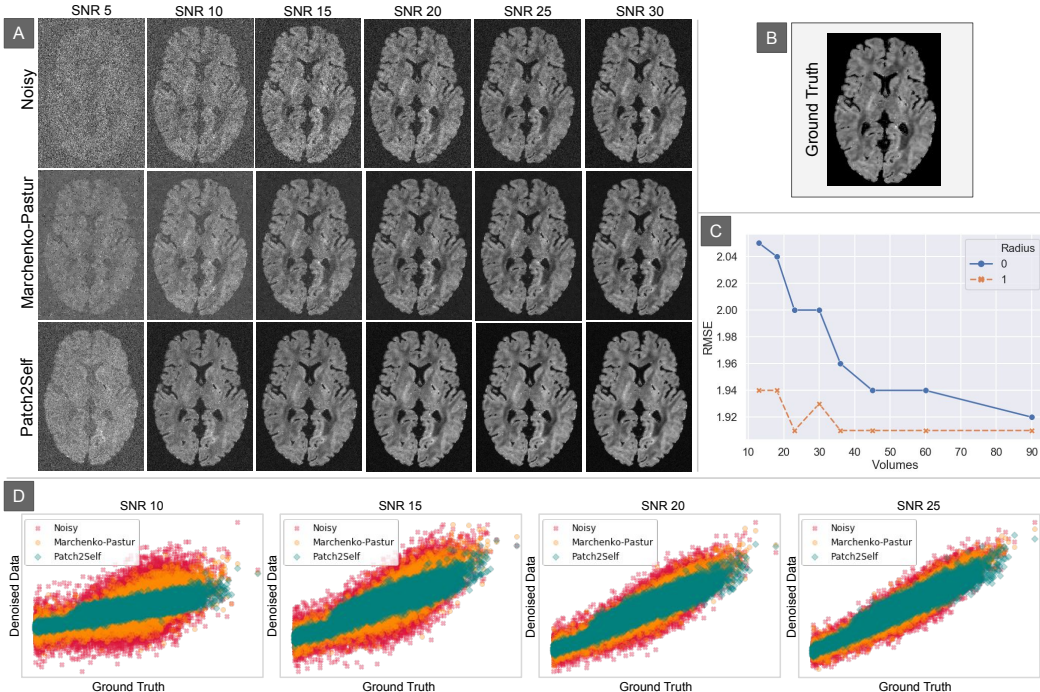

Figure 6: **(A)** Qualitative comparison of denoising on simulated data with varying levels of noise (from SNR 5 to 30) [see Sec. 4 for details of simulation]. The top row shows a slice of the simulated noisy data, the middle and the bottom row correspond to the Marchenko-Pastur and Patch2Self denoised outputs, respectively. **(B)** Plots the ground truth for the same slice. **(C)** Shows line plots of RMSE values at different number of volumes at patch radii 0 and 1. **(D)** Scatter plots of ground truth against denoised data of the simulated phantom at SNR 10, 15, 20 and 20. In each case, Patch2Self suppresses more noise and shows a consistent performance gain as the SNR increases (see table 1).

| | SNR 5 | | SNR 10 | | SNR 15 | | SNR 20 | | SNR 25 | | SNR 30 | |
|---|---|---|---|---|---|---|---|---|---|---|---|---|
| | $R^2$ | RMSE | $R^2$ | RMSE | $R^2$ | RMSE | $R^2$ | RMSE | $R^2$ | RMSE | $R^2$ | RMSE |
| Noisy | 0.04 | 14.84 | 0.27 | 5.57 | 0.52 | 3.05 | 0.69 | 2.02 | 0.79 | 1.47 | 0.85 | 1.15 |
| Marchenko-Pastur | 0.10 | 14.40 | 0.52 | 5.27 | 0.73 | 2.79 | 0.84 | 1.79 | 0.88 | 1.28 | 0.91 | 0.98 |
| Patch2Self | 0.20 | 13.86 | 0.69 | 5.18 | 0.84 | 2.74 | 0.89 | 1.73 | 0.91 | 1.25 | 0.93 | 0.98 |

Table 1: Reports the $R^2$ and the Root Mean Squared Error (RMSE) metrics on the simulated data at SNRs 5 to 30. The metrics have been computed for noisy (simulated phantom) data, Marchenko-Pastur and Patch2Self denoised data by comparing against ground-truth (noise-free) data.

## 5  Conclusions

This paper proposes a new method for denoising Diffusion Weighted Magnetic Resonance Imaging data, which is usually acquired at a low SNR, for the purpose of improving microstructure modeling, tractography, and other downstream tasks. We demonstrated that denoising by Patch2Self outperforms the state-of-the-art random-matrix-theory-based Marchenko-Pastur method on these subsequent analyses. To enable broad adoption of this method by the MRI community, we will incorporate an efficient and unit-tested implementation of Patch2Self into the widely-used open-source library DIPY.

## Broader Impacts

The broader impacts of this work fall into three categories: the direct impact on medical imaging, the theoretical impact on self-supervised learning more broadly, and the societal impact of improvements to those two technologies. In medical imaging, better denoising allows for higher quality images with fewer or shorter acquisitions, potentially making advanced acquisition schemes clinically viable, allowing for new bio-markers, and visualizing small structures such as the spinal cord in MRI.

Patch2Self provides a method for doing fast local matrix approximations, which could be used for matrix completion, subspace tracking, and subspace clustering, with applications across signal processing domain. To the extent that self-supervision enhances the ability to extract signal from poor measurements, it may expand the reach of state or private surveillance apparatuses allowing people's identities, movements, or disease status to be obtained from a greater distance and at lower cost. If a cache of easily acquired low-quality data can be efficiently used, it may open the door to exploitation by new actors.

## Acknowledgments and Disclosure of Funding

We sincerely thank Prof. Qiuting Wen (Indiana University School of Medicine) for providing the simulated data used in the above experiment. S.F. and E.G. were supported by the National Institute of Biomedical Imaging and Bioengineering (NIBIB) of the National Institutes of Health (NIH) under Award Number R01EB027585. J.B. was supported by the Chan Zuckerberg Biohub.

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
