[Supplementary Material]

# Patch2Self: Supplement

**Shreyas Fadnavis**[1][*], **Joshua Batson**[2][†], **Eleftherios Garyfallidis**[1][†]
[1]Indiana University Bloomington, [2]CZ Biohub

## 1 Regression Model Selection: Self Supervised Loss

As mentioned, out code-base allows for switching between any regression model from [8]. In order to choose a model, one can compute the self-supervised loss described in the main document:

$$\mathcal{L}(\Phi_J) = \mathbb{E}\|\Phi_J(X_{*,*,*,j}) - \hat{X}_{*,*,*,j}\|^2 \tag{1}$$

This self-supervised loss is basically the error between the noisy volume $(X_{*,*,*,j})$ and the denoised output for that volume $(\hat{X}_{*,*,*,j})$. Thus we can compute the self-supervised loss for each 3D volume of the 4D DWI data. The metric used to compute the self-supervised loss was the Mean Squared

Figure 1: (A) Shows the comparison between different types of regression models on the Stanford HARDI dataset [9]. Note that the outputs of each model, Ordinary Least Squares (OLS), Ridge, Lasso and Multilayer Perceptron are very similar. (B) Plots the self supervised loss for each volume in the data. Note that the Ridge and OLS give very similar values for the loss and Lasso performs gives a higher loss value, indicating relatively poor performance.

Error. From Fig. 1, we can see that OLS and Ridge give very similar performance and the Lasso performs slightly worse. The [9] data set has 160 volumes and the loss was computed for each one of them. For both Ridge and Lasso, the regularization parameter *alpha* was set to '1.0'. The cyclic feature selection method was used in the case of Lasso. One can even try to use a nonlinear regression function such as the Multilayer Perceptron/ Fully Connected Networks. To demonstrate the results shown in Fig. 1, we made use of 2 fully connected layers with 100 hidden units each. A linear activation function was used within each hidden unit. The adam [3] optimizer was used with a learning rate of 0.001 and $l_2$ penalty with $\alpha = 0.0001$ was used.

But these models come with their own set of hyperparameters which may require tuning according to the dataset at hand. The time required for fitting non-linear models is also higher, suggesting linear models are well suited for this type of a task. This setup is also supported by the previous approaches to DWI denoising [5, 10], where a local Singular Value Decomposition (SVD), essentially a linear model gave really good results.

## 2 Comparison with more methods

Here we compare Patch2Self with more methods apart which have been broadly adopted by the DWI community. These include the Adaptive Optimized Non-Local Means (AONLM) [7], Overcomplete Local PCA (LPCA) [5] (from [1]) and Marchenko Pastur PCA [10].

Figure 2: (A) Compares the denoised outputs of Overcomplete-LPCA at different ranks from 1-13 (with an increment of 3) (B), (C) and (D) Depict the denoised outputs and their corresponding residuals from the AONLM, Marchenko-Pastur and Patch2Self respectively. Each of them have been computed using the Sherbrooke 3-Shell dataset (slice 30, volume 44).

As shown in Fig. 2, we show the effect of different empirical thresholds on the rank of the local SVD using the LPCA algorithm [5]. They also suggest using the noise modeling framework [4] with an empirical threshold of a factor (2.3) of the standard deviation to decide upon the local rank. One can see that if the rank is too low (ranks: 1 to 4), a lot of structure is lost and the image has a lot of induced smoothness. If the rank is too high, a lot of noise starts getting into the image, reducing the performance of the denoiser. Similarly, a lot of the signal is lost/ smoothed out in the AONLM approach. Marchenko-Pastur, which aims to automatically find a threshold for each local SVD [10], does improve the SNR, but also retains some noise. Patch2Self on the other hand finds the optimal representation with the proposed self-supervised regression, giving results similar to ranks: 7 to 10 of

the LPCA method. While Patch2Self is not forced to be low-rank like the LPCA based methods, it still is able to achieve similar performance in a completely unsupervised setting.

# 3 Simulated Phantom

Figure 3: Compares the denoising of Patch2Self against Adaptive Optimized Non-Local Means and Marchenko-Pastur PCA. The ground truth image of has been depicted on the right.

In Fig. 3, We also compare performances of the automatic denoising algorithms AONLM and Marchenko-Pastur with Patch2Self. Notice that AONLM tends to smooth out a lot of the structure in the DWI signal leading to loss of information. Marchenko-Pastur is relatively non-aggressive but also leaves in some noise. Patch2Self on the other hand looks very similar to the ground truth right from SNR 10.

# 4 Code and Data

## 4.1 Code

We provide a purely pythonic implementation of Patch2Self with properly documented functions following object-oriented programming principles (available in `patch2self/model`). All the figures generated in the paper and the supplement are reproducible as shown in tutorial notebooks (available in `patch2self/notebooks`).

Following software were used to compare Patch2Self on real and simulated data (each are the original implementations of the algorithms):

- The implementation of Marchenko-Pastur PCA used for comparison was done via the Mrtrix software package (C++) mentioned in [10]:
  Link: https://mrtrix.readthedocs.io/en/0.3.16/tutorials/denoising.html

- The implementation of the AONLM [7] algorithm was done using the Mat-lab package: https://sites.google.com/site/pierrickcoupe/softwares/denoising-for-medical-imaging/mri-denoising/mri-denoising-software

Using the above software implementations, we provide the denoised outputs of each in the data attached herewith.

**LINK TO CODE:** https://figshare.com/s/431cbe040e6986a1a316

## 4.2 Data

We have made use of different open-sourced data [2], [9], [1] and [6]. Each of them have been provided either via an anonymous link to the data or via the data fetchers available in DIPY [1]. The raw and denoised outputs from the software mentioned in Sec. 4.1 have been provided. One can reproduce the results easily by using the implementations. Results generated from Patch2Self can be reproduced as shown in the tutorial jupyter notebooks.

**LINK TO DATA:** https://figshare.com/s/0b7a1a11612db6934543