[Reviews · NeurIPS 2020]

Review 1

Summary and Contributions: This work investigates the use of a self-supervised task (Patch2Self) for DWI denoising. The task consists in reconstruction missing directions from held-out voxels. The methods therefore exploits the 4D nature of DWI data. By using left-out directions in the prediction the noise of the target is uncorrelated which allows the theory of the Patch2Self to apply.

Strengths: The paper has limited novel contribution in ML but it is well written, illustrated and comes with convincing experiments. Python code is also provided.

Weaknesses: I would simply suggest that alternative denoising strategies based on dictionary learning have been proposed for DWI images such as https://pubmed.ncbi.nlm.nih.gov/24084469/ https://www.ncbi.nlm.nih.gov/pmc/articles/PMC4679293/ Sparsity is an alternative to low rank structures and the issue of heteroscedastic noise is also mentioned in these works via a whitening step. I would suggest to comment on these and potentially compare these approaches with the proposed one. Very little details are provided on how the models are trained. Is it done with an online algorithm? More details would have been appreciated.

Correctness: Correct

Clarity: Clear

Relation to Prior Work: Pretty good although as pointed out some references are missing.

Reproducibility: Yes

Additional Feedback:


Review 2

Summary and Contributions: The paper proposes a patch-based denoising algorithm for diffusion-weighted MRI. It is essentially a patch-regression approach: each component image of a dwMRI acquisition (which usually contains from a few tens to a few hundred separate grayscale imaging each with slightly different contrast) is denoised via a linear patch-model learned from all the other images. The authors compare the approach with the Marchenko-Pastur algorithm, which they identify as state of the art (not sure how true that is, as it does not appear to be standard or even commonly used in diffusion-weighted MRI, yet it does appear a sensible choice of baseline). Simulation results suggest some benefits of the proposed approach of the baseline. Results on brain data sets show qualitatively less noisy maps and output of tractography.

Strengths: It is an important problem. Diffusion weighted images often have very low signal to noise and noise reduction can certainly help in estimating subtle parameters from them and in reducing perturbation for connectivity mapping. The approach seems sensible and novel. Others have used non-local means and self-similarity for noise reduction in diffusion-weighted MRI (Wiest-Daessle et al MICCAI 2008; Manjon JMRI 2010, MedIA 2012) as well as patch-based learning approaches for more general enhancement (Coupe et al NIMG 2013; Alexander MICCAI 2014, NIMG 2017; Tanno MICCAI 2017). Some discussion of these approaches and how they differ to what is proposed would be useful, but I do believe they are not quite the same as what is proposed.

Weaknesses: Comparison with baseline not fair. One key difference between the proposed patch-based approach and the baseline Marchenko-Pastur is the patch-based nature of the former. Is smoother appearance of the images in figure 2 and the less noisy tractograms in figure 3 simply because the patch-based approach introduces more smoothing? That seems very likely to me. A potential big advantage of the Marchenko-Pastur is that it does not smooth so preserves detail. This is not tested in the qualitative evaluations of figures 2 and 3 or mentioned anywhere in the text. Experiments not very meaningful. As mentioned above, the qualitative comparison in figure 2 is hard to make judgement on. The new images are certainly smoother, but does that really mean “better”. The experiment would benefit from independent assessment from experts on both noise level and level of detail. Similarly the tractography result certainly shows a less noisy appearance, but is important detail lost? As for the kurtosis maps, I’m not sure the results make sense at all. Kurtosis mapping is a simple linear estimation problem. It is unclear what degeneracies the authors refer to in figure 4 that lead to the band of dark pixels. The dark pixels are consistent so it doesn't seem like the result of a degeneracy causing a broad set parameter values that offer equally good fit. The proposed approach certainly produces something different, but not clear it is better in any way. Niche application that does not generalise. While this work may prove important for the MRI community, it is unclear what the NeurIPS community would usefully learn. It seems like a bespoke solution for a specific application rather than a technique that potentially informs a wide range of machine learning problems – at least the authors give no clues as to wider applicability.

Correctness: As far as I can tell.

Clarity: Yes reasonably, although there is a lot of jargon relating to both diffusion MRI and noise reduction that is not easy to cut through or verify (for me at least).

Relation to Prior Work: Not particularly – see comments above.

Reproducibility: Yes

Additional Feedback: I appreciate the authors' rebuttal, which is reasonable on the points I raise, although I remain unconvinced on two points: Smoothness: the proposed method clearly adds more smoothing than the baseline approach. The authors accept that doesn't necessarily mean better. They argue that the metrics they provide account for that, but I don't see that they do. This seems an outstanding issue for me. Also still unconvinced on relevant: does this paper advance or provide insight more generally on the machine learning aspects of the work? Not particularly I don't think. That said, I am not strongly opposed to acceptance if others are more keen.


Review 3

Summary and Contributions: I have read the author response, read the other reviews, and participated in the discussion with the reviewers and area chair. My score is unchanged. The authors introduce a method for denoising DWI data with very few assumptions about the noise. The method is based on predicting voxel values in one direction from the values of those voxels + surrounding patches in all the other directions. This framework is general enough to support any regression model (as implemented in scikit-learn), though the authors use a simple linear regression. The authors compare their method against the state-of-the-art approach, and evaluate its effect on simulated and standard DWI data, and also on downstream analyses such as tractography and microstructure modelling. The authors state that the code will be released as part of DIPY, a commonly used library for handling diffusion data.

Strengths: strengths: - clearly written - the method is conceptually elegant and simple to understand/implement - evaluation on downstream analyses, not just DWI data

Weaknesses: nothing major

Correctness: The method is very well described, all my questions are in the Clarity section.

Clarity: 2.2) The process you describe in "Extracting 3D patches" uses data from all directions but one in predicting the left out direction j, for each direction in turn. This would be part A in Figure 1, and m = h * l * w. What is the new data that would be used in part B? Or do you mean you will literally replace each direction by its patch-wise prediction from the other directions? 139-144: is it reasonable to assume that images from different directions are different enough that you would have no collinearity in a regression model? (of course you can use regularization, if that's the case) 150: presumably this will also be the case if the dimensionality of the volume increases? 3) Figure 2: it's not really clear that there is any visual difference between one method and the other, in each dataset. Is there some quantitative criterion you can use, in the absence of ground truth?

Relation to Prior Work: Yes, the description of prior work appears very thorough, to the limited extent I know about this literature.

Reproducibility: Yes

Additional Feedback:


Review 4

Summary and Contributions: After the rebuttal and discussion, my opinion is still that this is a nice simple idea that seems to work well in practice, but I am also still a bit concerned about the insight that it provides in ML aspects, as well as the validation. ********** This paper presents a general method for denoising diffusion MR in an unsupervised fashion, where one learns to predict one diffusion weighted volume (channel/direction) from the other directions, and replaces the noisy measurement by the prediction. The idea is that, since the noise is independent across channels, predicting the noise is not possible and the method thus yields an unbiased estimator.

Strengths: -Simple method that seems to do a good job. -Good effort to validate on synthetic and real data, both with noise metrics, and on downstream tasks (e.g., tractography).

Weaknesses: -The validation on real data is largely qualitative (including the tractography, where - I totally understand - ground truth is very hard to obtain), and the quantitative part is very indirect (R2, degenerate voxels).

Correctness: Yes. It's a simple method, and it is well described.

Clarity: Yes, the paper is very clearly written.

Relation to Prior Work: The baseline is a good representative method for unsupervised DWI denoising. However, some classical denoising methods are not acknowleged, including (but not limited to): XQ-NLM Lam, Baban, et al Awate & Whitaker Tristan-Vega et al

Reproducibility: Yes

Additional Feedback: -If I understood correctly: the measured data in a given direction is not used (not directly) to estimate the denoised value. This is pretty interesting and I believe the authors could discuss it further in the manuscript. - The authors should better justify the use of [43] as primary baseline. - Comparison with supervised methods (as a ceiling for denoising performance would be informative. - Another discussion point: if a very flexible regressor is used (e.g., a very deep neural net), is it possible that it overfits and the method doesn't work? May that be why the linear model performs the best?

[Author Response · NeurIPS 2020]

We thank the reviewers for their positive comments on clarity, novelty, and convincing experiments. As R3 noted, one
key feature of the method is that the measured data in a given direction is not directly used to estimate the denoised
value: we literally replace each direction with its patch-wise prediction from the other directions using a linear model.
It was unexpected to us too that such a simple method would work so well. This is a key advantage of self-supervised
learning for medical imaging applications such as Diffusion-Weighted MRI (DWI).

We thank **R1** for pointing to the resources for denoising using dictionary learning and we will cite the mentioned
papers. Those papers have two limitations: they are designed for densely sampled and long acquisitions (DSI) and also
require learning the dictionary on high resolution data. Patch2Self in contrast works on any acquisition scheme and is
unsupervised. Sparsity in a learned basis is an important approach distinct from our own, and we will mention it. The
model is not trained in an online manner. We train one regressor per held-out volume (§2.2). To further clarify the
training, we have included a descriptive implementation with examples on open source datasets.

The comments from the **R2** are constructive and we address the criticism herewith. Both Patch2Self and Marchenko-
Pastur (MP) are patch based algorithms, assembling voxels from a patch across volumes into a matrix. MP uses
a low-rank approximation of that matrix, with thresholds based on random matrix theory which depends on an
assumption of homoskedastic noise. By learning a self-supervised regressor, Patch2Self relies on a weaker assumption
of independent noise over different volumes and can learn a full rank model. We agree with R2 that smoother does not
imply better, and therefore have provided extensive qualitative and quantitative comparisons on both synthetic and real
data. We also agree that the qualitative comparisons are hard to make and therefore provide quantitative comparisons
using goodness-of-fit measure for tensor and spherical harmonic models that downstream analyses depend on. Improved
RMSE and $R^2$ scores on realistic synthetic data have been reported as more direct measures of performance. We would
like to note that the tractography comparisons are in fact quantitative in nature, where a kernel density estimate is
performed on each streamline to evaluate the spurious streamlines from the tracking (using the Fiber Bundle Coherency
(FBC) Metric) and the number of streamlines is counted. R2 is correct that diffusion kurtosis (DKI) is a linear estimation
problem. By degeneracy, we refer to obtaining a biophysically impossible parameter estimates. The microstructure
model, here DKI, is unable to fit due to the low SNR of the acquired data. DKI is also a very widely used higher-order
model following the initially proposed Diffusion Tensor (DTI) mapping, motivating us to include it as a part of our
analysis. In the presence of more noise, one can imagine that the slope of the line fitted to the DWI data to be reversed,
causing a degeneracy (seen as dark black voxels). Values close to zero are not plausible in a healthy brain and can
therefore confound DKI analyses, classifying that voxel as disease/ abnormal (as in the case of tissue degeneration).
As pointed out by the arrows, the number of these degenerate (black) voxels have been reduced in the data. We do
agree that there are some more degeneracies that the denoising could not suppress, but may be indicative of other
artefacts such as Gibbs oscillations/ ringing. With respect to impact, the paper is aimed for the brain imaging category
of NeurIPS. DWI is currently the most powerful non-invasive way of assessing structural information, and denoising is
an essential component of DWI analysis. The proposed approach of self-supervised learning can also be extended to
other 4D modalities like fMRI.

We thank the **R3** for the positive feedback. The reviewer is correct in identifying part A, where patches corresponding
to the direction j are held-out and used as targets for training a regression function $\Phi$. The rest of the directions were
used as features, and the model is trained on all voxels. In part B, the same training samples are given to the trained
regressor, whose output is the denoised volume. The rationale behind doing so is that, since the noise across volumes
is uncorrelated, the regressor will only be able to learn to predict the underlying signal and not the noise component.
Collinearity is unlikely in DWI because of noise in the measurements. Moreover, the similar performance of OLS, L1-
and L2- regularized regression implies that collinearity is not a problem (§S1). As for the comment on line line 150 by
R3, an increase in dimensionality per volume will only increase the number of training samples, not the number of
features, per volume (giving more information about the same object). Only an increase in the number of volumes gives
the regressor access to different information about the held out volume. For Fig. 2, although direct comparisons cannot
be done on real data, we provide a way to quantify the improvements due to denoising using $R^2$ and FBC measures on
downstream tasks of microstructure analysis and tractography.

While we do agree with **R4** that the $R^2$ metric is indirect, we also provide the more direct RMSE scores for the synthetic
phantom in Table §1. (This is not possible in the case of the real data due to the absence of ground truth.) We thank
the reviewer for pointing out the mentioned references and will be included in the final document. For a justification
of using MP as a baseline for comparison, we would like to point out that MP is the most cited method for DWI
denoising (320+ citations since 2016) and is well received by the community from an application standpoint. Open
source implementations are available in popular software packages such as MRtrix3 and DIPY. Without ground-truth
for these datasets we are unable to train supervised methods, but we do compare against the state-of-the-art classical
DWI algorithms such as AONLM and Local PCA in the supplement (see §S2 and §S3). The suggestion to compare to a
deep neural net is useful. Our expectation is that it will not do much better because a shallow fully connected neural
network (§S1) had a higher loss than the linear model.

[Meta-Review · NeurIPS 2020]

Despite the novelty of the proposed method might be considered marginal with respect to the machine learning community, the contribution to the application field is relevant. The availability of the code represents an added value in the perspective of open science. The authors provided satisfactorily answers in the rebuttal.